# Fabrication Mechanisms of Lignin Nanoparticles and Their Ultraviolet Protection Ability in PVA Composite Film

**DOI:** 10.3390/polym14194196

**Published:** 2022-10-06

**Authors:** Jiawei Zhang, Zhongjian Tian, Xingxiang Ji, Fengshan Zhang

**Affiliations:** 1State Key Laboratory of Bio-based Materials and Green Papermaking, Qilu University of Technology (Shan-dong Academy of Sciences), Ji’nan 250353, China; 2Shandong Huatai Paper Co., Ltd., Dongying 257335, China

**Keywords:** lignin, PVA-polymerized film, UV-resistant, self-assembly, nanoparticles

## Abstract

Lignin is an indispensable and essential compound present in plants. It is a renewable resource and a green alternative to traditional petroleum energy. The rational utilization of lignin can reduce the environmental damage caused by traditional industrial development. The preparation of lignin nanoparticles (LNPs) using the self-assembly method is one of the most favorable ways to achieve high value-added utilization of lignin. However, the process requires an in-depth understanding of the sphere-forming mechanism of lignin self-assembly and the interaction of self-assembly forces. We used the same raw materials and two different preparation methods to prepare LNPs. The results revealed that the variation in the order of the dropwise addition of lignin solution and deionized water produced LNPs with varying average sizes. The sphere-forming mechanisms of the two kinds of lignin nanoparticles were discussed for the preparation of UV-resistant polyvinyl alcohol (PVA) polymeric films. During lignin spherification, the faster the solution reaches the supersaturation state, the faster the spherogenesis rate is, the smaller the size is, and the narrower the particle size distribution is. The lignin micro/nanospheres are produced by exploiting the π–π bonding interactions in lignin itself. The lignin micro/nanospheres are then mixed with PVA to form a film to obtain a lignin–PVA composite film material with an anti-UV effect.

## 1. Introduction

Today, the world is concerned about the energy crisis and climate change. This concern has resulted in the global development of high-value-added renewable biomass resources. Lignin is an abundant aromatic macromolecule present on earth. This is a natural polymer with a complex structure formed by dehydrogenation and polymerization of lignin monomers with different degrees of methoxylation. It is primarily composed of three monomers, guaiacyl (G, containing one methoxy), syringyl (S, containing two methoxys), and p-hydroxyphenyl (H, without methoxy), connected by C–C and C–O bonds. The molecule has broad application in various areas and good prospects [1,2,3]. Lignin is found in the cell wall of plants and plays a vital role in the development of plants. It promotes vertical growth, helps in photosynthesis, and ensures plants’ normal water transport. It also resists the effects of ultraviolet (UV) light and biological erosion [4,5,6,7]. In terrestrial plants, lignin is the second most abundant biomolecule after cellulose, accounting for approximately 30% of the total organic carbon in living organisms. The alkali pulp and paper industry produces a large amount of black liquor every year, most of which is burned directly. A small portion of the compound is extracted from kraft lignin and used at a low value. It is used as a burning fuel that pollutes the environment. In addition, effective utilization of lignin and waste lignin resources cannot be realized [8,9]. The structure of kraft lignin (KL) is very different from that of natural lignin, as a large amount of the original structure is destroyed during the pulping process. This results in the subsequent derivation of various new complex structures. KL changes with the change in the biomass raw materials and the production processes, which limits the efficient use of KL [10,11,12]. Therefore, the realization of high value-added utilization of lignin is the main driving force in utilizing renewable biomass resources [13]. KL, obtained as a cheap byproduct in the pulp and paper industry, is available in large quantities [14]. The manufacturing costs for nanoparticles obtained from lignin are estimated to be approximately USD 1000 per ton [15].

Lignin nanoparticles (LNPs) are currently one of the materials used for the effective high value-added utilization of lignin [16,17,18]. As LNPs are characterized by high specific surface areas and densities, as well as high surface permeability and colorability, the development of new nanolignin-based material can help in the high value-added utilization of lignin. This has attracted much attention as these materials can be used as drug carriers, antibacterial agents, anti-UV materials, emulsion stabilizers, and electrocatalytic carbon materials [19,20,21,22,23,24,25,26]. To date, LNPs have been primarily obtained following self-assembly (anti-solvent) [27,28,29,30,31], emulsion solvent evaporation [32,33], pH conversion [34,35,36], electrostatic spinning [37,38], physical grinding [39] and other processing methods [40,41,42,43,44]. Jiang et al. [27] used 90%, 70%, and 40% ethanol to fractionate the sulfated lignin into three fractions, labelled as LS90, LS70, and LS40. Among them, lignin nanorods were prepared from LS40, and nanospheres were prepared from LS70 following the anti-solvent method. Wang et al. [28] first fractionated the lignin systems following the solvent extraction process. Subsequently, the optimal components were prepared to form homogeneous, monodispersed LNPs exploiting solvent/anti-solvent self-assembly properties to improve the homogeneity of the intermolecular forces in lignin during the self-assembly process. Liu et al. [29] prepared LNPs by adding an anti-solvent of water. The effective dissolution of industrial alkali lignin in imidazole ionic liquid was realized. Trevisan et al. [30] isolated pure lignin from elephant grass using two simple extraction methods in the presence of dilute acid and alkali solutions. The lignin acetate nanoparticles were prepared using acetone as the solvent and water as the anti-solvent. Chen et al. [31] prepared LNPs following the self-assembly method using dimethyl sulfoxide as the solvent and water as the counter-solvent. Wang et al. [32] used a binary system consisting of green γ-valerolactone (GVL) and glycerol to prepare LNPs without additional lignin modification and dialysis. The basic principle of LNP formation lies in emulsifying lignin-containing GVL droplets in glycerol. The materials are first heated and then cooled. Sameni et al. [33] prepared emulsions by mixing lignin and ethyl acetate (EA) as the organic phase and polyvinyl alcohol as the aqueous phase. They added deionized water to the emulsions to form the LNPs. Liu et al. [34] prepared LNPs by dissolving lignin in an ionic liquid [Emim][AC], adding deionized water to the system, and adjusting the pH to 2–3. Cha et al. [35] extracted polydisperse lignin microspheres with homogeneous morphology directly from the black liquor by lowering the pH of the liquor to < 4. This was followed by hydrothermal treatment. Ma et al. [36] diluted the black liquor and made the lignin acid sink into the black liquor to form LNPs by lowering the pH of the liquor. Bahi et al. [37] prepared lignin nanofibers following the electrostatic spinning method. Camiré et al. [38] prepared lignin nanofibers from alkali lignin by electrostatic spinning and mixed them with co-polymer poly (vinyl alcohol) to prepare lignin nanofiber films. Mousavi et al. [39] extracted the lignin from the alkali pulping black liquor. The size of the lignin particle was reduced to the micron and nanometer levels under conditions of physical grinding. Other methods were described by Chen et al. [40], Liu et al. [41], He et al. [42], Taverna et al. [43] and Agustin et al. [44]. Different preparation methods are used to prepare lignin nanoparticles that find their utility in different fields. We have investigated the previously reported preparation methods and studied the characterization methods used to determine the morphology and properties of the systems.

The conventional self-assembly method is to form nanoparticles by gradually adding the anti-solvent dropwise to the solvent. The large particle size and irregular morphology of LNPs limit the performance of lignin as a high value-added material. Recent studies have found that overcoming the inherent drawbacks of lignin, such as inhomogeneity, large particle size, and low yield, becomes a significant challenge to lignin nanoparticles. We aim to develop relatively simple, economical, environmentally-friendly, and high-yield LNPs. The traditional self-assembly method involves the addition of the configured lignin solution slowly and dropwise into the counter-solvent. The sample is obtained under conditions of dialysis, centrifugation, and freeze-drying. In our work, we prepared and characterized the LNP samples by adding the solvent dropwise to the ani-solvent. This method solves the disadvantages of uneven size distribution and poor stability of lignin nanoparticles, and a simple and efficient way for preparing lignin nanoparticles is provided. We labelled the sample produced following the conventional anti-solvent method (CLMPs) as the lignin microspheres. This sample is referred to as the LNPs prepared following the trans anti-solvent method (TLNPs).

## 2. Materials and Methods

### 2.1. Chemicals and Materials

Kraft lignin (KL) was provided by Rizhao Huatai Paper Co., Ltd. (Shangdong, China) and was used with further purification. Tetrahydrofuran (THF, ≥99.8%), 1,4-dioxane (≥99.5%), NaHCO_3_ (NaHCO3, ≥99.5%), hydrochloric acid (HCl, 36.0~38.0%), and potassium bromide (KBr) were purchased from Sinopharm Chemical Reagent Co., Ltd. (Shanghai, China). The dialysis bags with a molecular weight cutoff of 1000 Da were purchased from Mym Biological Technology Co., Ltd., and PVA from Beijing Yili Fine Chemicals Co., Ltd. (Beijing, China).

### 2.2. Purification of Kraft Lignin

KL (200 g) and 1000 mL of dioxane/1,4-dioxane water (18.25 MΩ·cm) acidic mixed solution (*v*/*v* = 9:1, pH = 2) were added into a 2500 mL three-necked flask. The mixture was stirred thoroughly to form a homogeneous solution. Subsequently, the flask was put into an oil bath at 87 °C to extract the lignin–carbohydrate complexes after 2 h. The process was conducted under a constant flow of N_2_. After collecting the supernatant by centrifugation, NaHCO_3_ (5.95 g) was added to the supernatant, and the mixture was stirred for 3 h to neutralize the additional acid completely. After removing the precipitate and 1,4-dioxane by centrifugation and rotary evaporation at 55 °C, dilute hydrochloric acid (pH = 2) was added into the resulting solution to promote the precipitation of lignin. Finally, the purified alkaline lignin was obtained by centrifugation. The product was washed thrice with dilute hydrochloric acid (pH = 2), and was then freeze-dried [45].

### 2.3. Preparation of Lignin Nanoparticles

The lignin nanospheres were prepared following a conventional/trans anti-solvent method, as shown in Figure 1a,b. The process of the formation of LNPs is presented in Figure 1. KL was first dissolved into 100 mL of THF to form a homogeneous solution (5.0 mg/mL) at ambient temperature. Subsequently, the prepared lignin solution was mixed with deionized water (600 mL) at a rate of 5 mL·min^−1^ and stirred at 600 rpm. The addition process was controlled with a peristaltic pump (YZ1515X-A, Baoding Longer Precision Pump Co., Ltd., Hebei, China). The solution was placed into four dialysis bags (200 D) and dipped into deionized water at different times (2, 4, 8, and 16 h) at 50 °C. The solution was replaced once every 2 h to remove THF rapidly. Finally, the LNP was obtained, followed by freeze-drying the dialyzed solution. The samples were marked as TLNPs-X (X = 2, 4, 8, and 16), where X stands for the number of hours of dialysis. For comparison, the lignin microspheres were synthesized by regulating the drip sequence for the TLNPs (i.e., deionized water was dripped into the lignin solution) (Figure 1b.). Accordingly, the resulting lignin microspheres were named CLMPs-Y (Y = 2, 4, 8, and 16).

### 2.4. Formation Mechanism of LNPs

Several factors influence the process of self-assembly that results in the formation of spheres. Rapid mixing of the solvent and anti-solvent is important for supersaturation. The degree of supersaturation of nanoprecipitation in solution is defined as follows:(1)$$S=C_{0}/C*$$
where *C*_0_ denotes the real-time concentration of lignin in the mixture of the organic solvent and anti-solvent, and *C** represents the saturation solubility of lignin in the mixed solution. The higher the supersaturation of lignin in the solution, the lower the systematic Gibbs free energy (Δ*G_max_*) of the solution. This results in a higher nucleation rate (B) that follows the Arrhenius relationship:(2)$$B=k_{1}\exp(-\frac{\Delta Gmax}{KT})$$

The relationship between the sphere formation rate and lignin concentration can be expressed as follows:(3)$$B=k_{n}{\left(C_{0}-C^{*}\right)}^{n}$$
where *k_n_* is the spherogenesis rate constant, and n is the kinetic index of the spherogenesis process. The relationship between the sphere-forming rate and supersaturation is obtained from Equations (1) and (3) as follows:(4)$$B=k_{n}C^{*}{}^{n}{\left(S-1\right)}^{n}$$

The ratio of τ_mix_ to τ_flash_ corresponds to a dimensionless Damkohler number [46] for precipitation (*D_ap_*) and is expressed as follows:(5)$$D_{ap}=\frac{\tau_{\mathrm{mix}}}{\tau_{\mathrm{flash}}}$$

The time parameters associated with the nanosphere formation process are mixing time (τ_mix_) and ball formation time (τ_flash_). These time scales significantly affect particle formation and are controlled by mixing. When *D_ap_* < 1, the mixing time of lignin solution and water is shorter than that required for the sphere-forming and growth phases. In this case, the supersaturation state is rapidly reached. The process of spherification occurs quickly, and it dominates the nanosphere process. A large number of small and homogeneous nanoparticles are formed under these conditions. When *D_ap_* > 1, the mixing time of solvent and anti-solvent (τ_mix_) is longer than that of the spherogenesis process (τ_flash_), and the critical spherogenic concentration of lignin is slowly reached. This results in dominant sphere growth and an increase in particle size [46].

### 2.5. Preparation of LNP/PVA Composite Film

PVA (5 g) was mixed with 95 g of deionized water, and the mixture was stirred at 95 °C for 2 h at 600 rpm to obtain a 5% PVA aqueous emulsion. Following this, aqueous solutions of TLNP and CLMP were added to the PVA aqueous emulsion under the same conditions, and the mixture was stirred for 1 h. The PVA polymeric films with mass fractions of 1%, 5%, and 10% for lignin micro-nanosphere solutions were formulated.

### 2.6. Material Characterization

The morphological analyses of TLNPs and CLMPs were performed using a field-emission scanning electron microscope (SEM, TESCAN MIRA LMS, Kohoutovice, Czech) and a transmission electron microscope (TEM, JEOL JEM 2100F, Tokyo, Japan). Based on the SEM images, the size distribution data of the LNP were collected using open-source software (Nano Measurer, Xi′an, China). The Fourier transform infrared spectroscopy (FTIR, Nicolet iS20, Thermo Fisher Scientific, Shanghai, China) technique was used to characterize the surface functional groups in the wavenumber region of 500–4000 cm^−1^. X-ray diffraction (XRD) patterns were recorded using a Rigaku RU-200B X-ray diffractometer (Tokyo, Japan) using a Cu Kα radiation source (40 kV, 30 mA). The surface element contents and species of the samples were characterized using the X-ray photoelectron spectroscopy (XPS, Thermo Scientific ESCALAB Xi+, Waltham, MA, USA) technique. A monochromatic Al Kα source was used, and the binding energies were corrected with a binding energy of C1s (284.5 eV). Thermogravimetric analysis (TGA) was performed using a TGA 550 thermal analyzer. The specimens were heated from 30 to 800 °C (atmosphere: N_2_, heating rate: 5 °C min^−1^). The dispersion stability of the TLNP and CLMP were qualitatively analyzed using the Turbiscan stability index (TSI) obtained with a Turbiscan Tower, France. UV-vis transmittance spectra were captured using a SHIMADZU UV-2600 UV-visible spectrophotometer (Kyoto, Japan) from 290 to 400 nm.

## 3. Results and Discussion

### 3.1. Morphologies and Characteristics of TLNPs and CLMPs

SEM images of TLNPs and CLMPs are illustrated in Figure 2. It was observed that the size of TLNP was significantly smaller than that of CLMP, and the size distribution of CLMP was not uniform. The size of the spheres was approximately distributed between 200 and 800 nm. In contrast, the size of TLNP was uniform and primarily distributed in the range of 40–80 nm. The surface of the prepared LNPs appeared smooth.

Figure 3 shows the particle size distribution of CLMPs and TLNPs and the comparison of the yield of lignin prepared following the two methods. Figure 3a–d. presents the particle size distribution of CLMP recorded under conditions of different dialysis times. Figure 3e–h shows the particle size distribution of TLNP recorded under conditions of varying dialysis times. It was observed that the particle size distribution was primarily in the range of 40–80 nm. The particle size distribution became more uniform and homogeneous with an increase in the dialysis time. Figure 3i presents the average size of the LNPs prepared following the two methods, and it was found that the size of the LNPs corresponding to CLMP was one order of magnitude higher than that of the LNPs corresponding to TLNP. Figure 3j presents the yields of the LNPs prepared following the two different methods. The comparison shows that the work of TLNP is stable above 90%, which is approximately three times higher than that of CLMP. The TSI values corresponding to TLNP and CLMP at the same concentrations were determined further to verify the dispersion stability of TLNP and CLMP. The TSI value of TLNP reached a plateau at approximately 22 h. The significantly high value reflected the stable dispersion of TLNP (Figure 3k). Figure 3l shows the sample images recorded after centrifugation at 4000 rpm for 10 min.

### 3.2. Structural Characterization of LNPs: FTIR, TG, and XPS Analysis

The FTIR spectral profiles are presented in Figure 4a, which illustrates the properties of kraft lignin, TLNP, and CLMP. According to reports in the literature, the absorption peaks at 2940 and 2850 cm^−1^ can be attributed to the C–H asymmetric and symmetric stretching vibrations of methyl CH_3_ and methylene CH_2_, respectively [47,48]. This indicates the presence of side chain structural units in lignin. The absorption peaks at 1710 and 1656 cm^−1^ in the carbonyl region originate from the absorption of the conjugated carbonyl group (C=O). The characteristic absorption peaks of the benzene ring skeleton appeared at 1515, 1600, 1426, 852 and 820 cm^−1^, indicating the presence of the benzene ring structure specific to lignin. In addition, the characteristic peak of the guaiacyl ring and the C–O stretching vibration appear at 1272 cm^−1^. The peak at 1221 cm^−1^ combines the absorption peak of the guaiacyl ring and the C–H vibration. The peak at 1154 cm^−1^ is the typical absorption peak representing the ether bond. This indicates the presence of kraft lignin. The absorption peaks at 1371 and 1216 cm^−1^ demonstrate the existence of phenolic hydroxyl groups. In summary, it can be inferred that both lignin samples are HGS-type. Relatively less variation was observed in the TLNP spectra compared to the CLMP spectra. This implies that the structure of the lignin molecules was not altered during self-assembly, which could be attributed to the bond dissociation energy [47,48,49]. Typically, lignin is composed of different kinds of aromatic rings and organic functional groups linked through chemical bonds. This results in the variation in the degradation temperature range of 25–800 °C. Three stages of degradation were observed for kraft lignin, CLMP, and TLNP samples (exception: dried biomass). Figure 4b presents the TGA curves of the sample. The initial weight loss at 25–150 °C was ascribed to the evaporation of absorbed moisture, while the gradual decline in weight observed between the temperature range of 150 and 600 °C was attributed to the degradation of lignin carbohydrates that result in the formation of volatiles such as CO, CO_2_, and CH_4_ [50]. The final degradation phase was slow and occurred at 600–800 °C. The surface functional groups of CLMP and TLNP were further investigated with the XPS technique. Four individual peaks were observed for the LNP sample in the C 1 s peak region (Figure 4c–f.). These peaks corresponded to the C−C or C−H (C-I, 284.8 eV), C−OH (C-II, 286.1 eV), O−C−O (C-III, 287.5 eV), and O-C=O (C-IV, 288.8 eV) groups. Compared to kraft lignin and CLMP, the C 1s spectrum of TLNP has a significantly weaker intensity of the O=C-OH peak at 288.5 eV. This can be attributed to the different formation mechanisms of CLMP and TLNP. Solid lignin microspheres (CLMP) are characterized by a hydrophilic outer surface and a hydrophobic core, while hollow LNPs (TLNP) are characterized by a hydrophobic outer surface and a hydrophilic inner surface. This indicates that the carboxyl content of lignin molecules has a significant effect on the hydrophilic and hydrophobic properties of the materials. These occurrences of the self-assembly reactions eliminated the chances of the presence of the carbon-containing functional groups, which subsequently reduced the carbon content and increased the oxygen content [51,52].

### 3.3. Formation Mechanism of LNPs

As shown in Figure 5, when the solvent (THF) is mixed rapidly with the anti-solvent (water), the lignin concentration increases to the saturation concentration (*C**). Subsequently, it reaches the critical sphere-forming concentration (*C*) that triggers the precipitation process. Compared to the sphere-forming process of CLMP, TLNP instantly reaches supersaturation during the sphere-forming process. This can be potentially attributed to the fact that a drop of lignin solution is added dropwise to a certain volume of water. This is equivalent to a certain volume of water being poured into a decreasing lignin solution at an infinite rate. Under these conditions, the conditions of supersaturation are instantly reached. At this stage, spheres form rapidly and continue to grow via lignin coalescence. The growth continues until a stable critical value is reached. When the solute concentration returns to the critical sphere concentration (*C*), new nucleation cannot occur. The existing nucleus continues to grow until the solute concentration drops to the saturation solubility (*C**), and this stage is labelled as the sphere growth phase. TLNP’s ball formation process is speedy and dominated by ball formation, while CLMP’s ball formation process is dominated by ball growth. A higher nucleation rate is required to obtain smaller nanospheres, and fewer nanospheres are grown. The rate of spherogenesis was found to increase with an increase in supersaturation. Throughout the spherogenesis process, spherogenesis and sphere growth co-occurred, and both processes competed for lignin supersaturation. If ball growth predominates in the supersaturated state, the final particles exhibit larger particle size and wider particle size distribution. Therefore, it is critical to enhance lignin spherification and inhibit ball growth during particle formation. 

In terms of lignin surface structure, Figure 6 presents the sphere formation mechanism diagram for CLMP and TLNP. Typically, chromato-graphic grade THF and water are mutually soluble in any ratio, and lignin is highly soluble in THF. Uniform THF–lignin dispersion is formed in solutions. This can be attributed to the fact that lignin contains many C–C and C–O bonds, and lignin exhibits a macromolecular network structure formed by the crosslinking of three primary monomers through C–C and C–O bonds. Kraft lignin contains a large number of structures similar to tetrahydrofuran oxetane. Thus, it can be well-dissolved in THF. Under these conditions, many phenolic hydroxyl groups appear in the dissolved lignin structure, increasing the reactivity of the lignin molecule [54,55]. When deionized water is added to the THF–lignin solution system during the process of preparing CLMP, the separation of phases occurs throughout the solution system, i.e., the continuous phase (THF) and the dispersed phase (water) separate [56]. As the lignin molecule is amphiphilic (contains hydrophobic benzene ring and hydrophilic phenolic hydroxyl), when the water content increases to a certain degree, the lignin molecule forms a film at the interface between the two phases of water and THF. This results in water being wrapped or encapsulated by water. A dynamic equilibrium between the continuous and dispersed phases is achieved [57]. As the deionized water content continues to increase, an increasing amount of water permeates the membrane, resulting in the self-assembly of numerous lignin molecules. The process occurs via the layer-by-layer method exploiting π–π interactions. Water is eventually collected at the inner surface of the membrane [58,59,60]. However, the increase in the water content results in an increase in the pressure gradient inside and outside the membrane, and the thinner side of the membrane breaks to keep the internal and external pressure stable [61,62]. The pressure difference between the inside and outside of the membrane results in the formation of pores on the outer shell of each hollow nanosphere. Incipient pores are formed in the thinnest part of the sphere surface wall. As shown in Figure 7c,d, a small pore is formed on the surface of the ball. This proves that the CLMPs are hollow core-shell structures. KL contains less methoxy groups and is characterized by a highly crosslinked structure [63]. When the lignin solution was added dropwise to deionized water, the lignin solution was rapidly dispersed in the water at the beginning of the process. Under these conditions, the lignin molecules rapidly form nanospheres in water, which can be attributed to their hydrophobic properties. The lignin molecules form small LNPs as the amount of lignin mixed with water is significantly small (Figure 7a,b). The driving forces of self-assembly include hydrogen bonding, van der Waals forces, electrostatic forces, and π–π bond interactions [64,65,66,67,68]. The driving force for the assembly of lignin molecules is considered to be the π–π interactions, and the self-assembly method is simple, convenient, and suitable for modern chemical formulations. Impurities are not introduced during the process. We hope this new method will contribute to developing new strategies for nanolignin materials with new structures and functions. The dissolution and aggregation of lignin can be well controlled by changing the order of the mutual dropwise addition of lignin solution and deionized water. This can potentially help to regulate the self-assembly of the solution.

### 3.4. Anti-UV Performance of Lignin–PVA Composite Film

The PVA polymeric films with mass fractions of 1%, 5%, and 10% for lignin micro-nanosphere solutions were formulated. As shown in Figure 8, the PVA film devoid of lignin does not exhibit anti-UV transmission properties, and as the content of the lignin solution is increased, the anti-UV transmission rate of the PVA polymeric film is effectively improved. The results were compared with those obtained by studying the same amount of PVA polymeric films containing lignin. It was found that the anti-UV effect of PVA-TLNP polymeric films was generally better than that of the PVA-CLMP polymeric films. This can be potentially attributed to the small particle size and narrow particle size distribution of TLNP.

## 4. Conclusions

Uniform lignin nanoparticles are obtained from KL, which comes from the purification of kraft lignin. In our work, LNPs were formed when KL-THF solutions were continuously added into water. This is different from the traditional anti-solvent method. The method solved the disadvantages of lignin nanoparticles, such as inhomogeneous size and poor stability. It was observed that not only did the size differ by one order of magnitude, but also the UV-resistance properties were also different. The smaller the size of the lignin spheres, the better the UV resistance. So, this method achieves no waste of biomass resources and high-value utilization of lignin. However, the functionalization of lignin at the nanoscale is still in the initial stage, and research on the development of lignin nanoparticles is gradually gaining momentum. Studies to achieve the high-value utilization of lignin are still nascent. On the one hand, composite materials are designed, and chemical modifications are made to improve the performance and application prospects of lignin. On the other hand, the fabrication of lignin nanoparticles of tunable size can help address the problems of social production. This could effectively result in the high value-added utilization of lignin.

## Figures and Tables

**Figure 1 polymers-14-04196-f001:**
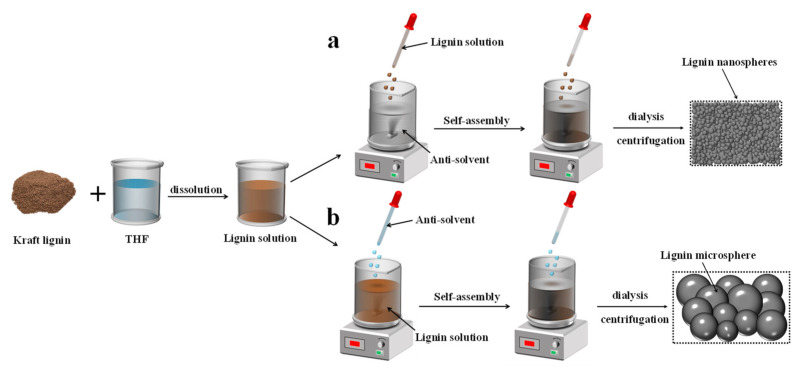
The synthetic route followed for developing lignin micro/nanospheres: (**a**) trans anti-solvent method for lignin nanospheres; (**b**) conventional anti-solvent method for lignin microspheres.

**Figure 2 polymers-14-04196-f002:**
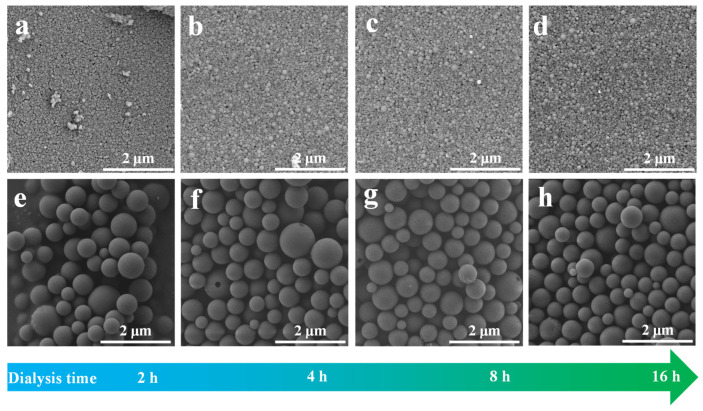
SEM images of (**a**–**d**) TLNPs-X and (**e**–**h**) CLMPs-Y, the first row of the image represents TLNP samples, and the second row represents CLMP samples, where X and Y represent dialysis time. (**a**–**d**) correspond to 2, 4, 8, 16 h respectively; (**e**–**h**) correspond to 2, 4, 8, 16 h respectively.

**Figure 3 polymers-14-04196-f003:**
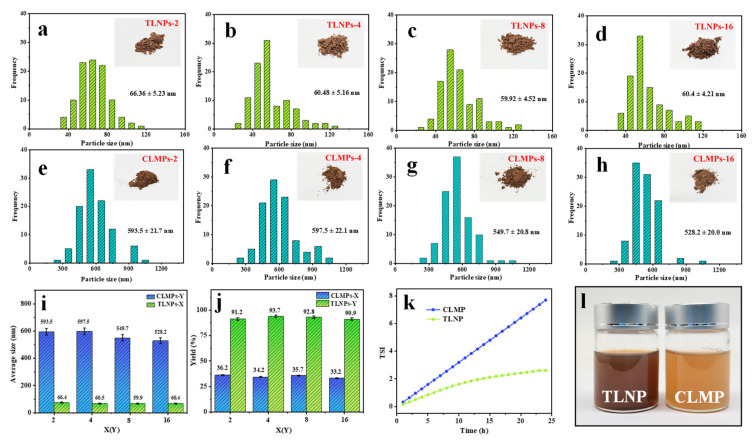
Size distribution curves recorded for TLNPs-X and CLMPs-Y: (**a**) X = 2, (**b**) X = 4, (**c**) X = 8, (**d**) X = 16, (**e**) Y = 2, (**f**) Y = 4, (**g**) Y = 8, and (**h**) Y = 16; (**i**) average sizes and (**j**) yields of TLNPs-X (X = 2, 4, 8, and 16) and CLMPs-Y (Y = 2, 4, 8, and 16); (**k**) dispersion stability of CLMP and TLNP; and (**l**) photograph of the centrifuged reaction solutions corresponding to TLNPs-16 and CLMPs-16.

**Figure 4 polymers-14-04196-f004:**
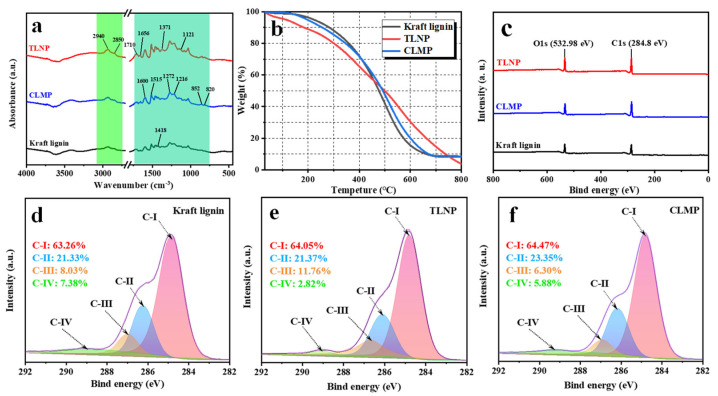
FTIR spectral profiles (**a**), TG (**b**), XPS (**c**), and C 1s spectra (**d**) kraft lignin, (**e**) TLNP, (**f**) CLMP recorded for kraft lignin, TLNP and CLMP.

**Figure 5 polymers-14-04196-f005:**
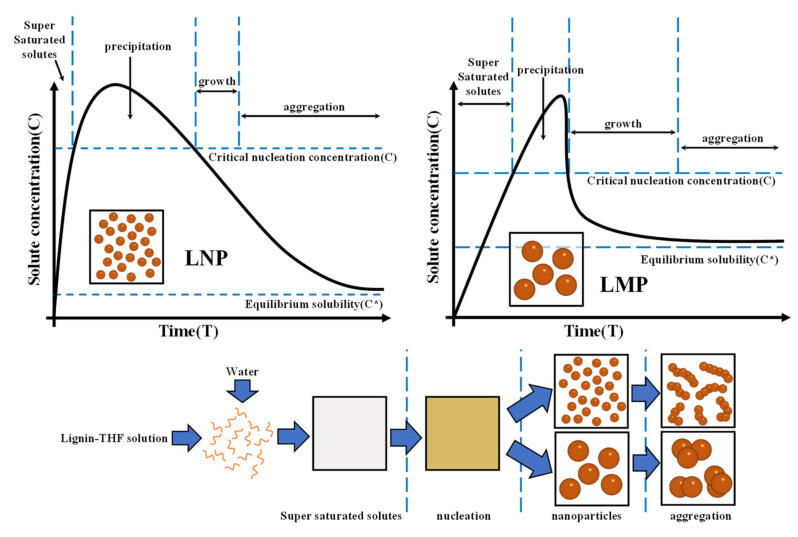
La Mer model [53] and schematic diagram of the lignin nanoparticle forming process.

**Figure 6 polymers-14-04196-f006:**
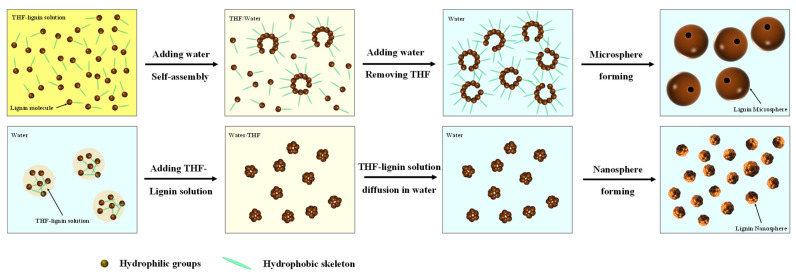
Lignin condensation facilitates solution-based self-assembly for the production of spherical lignin nanoparticles.

**Figure 7 polymers-14-04196-f007:**
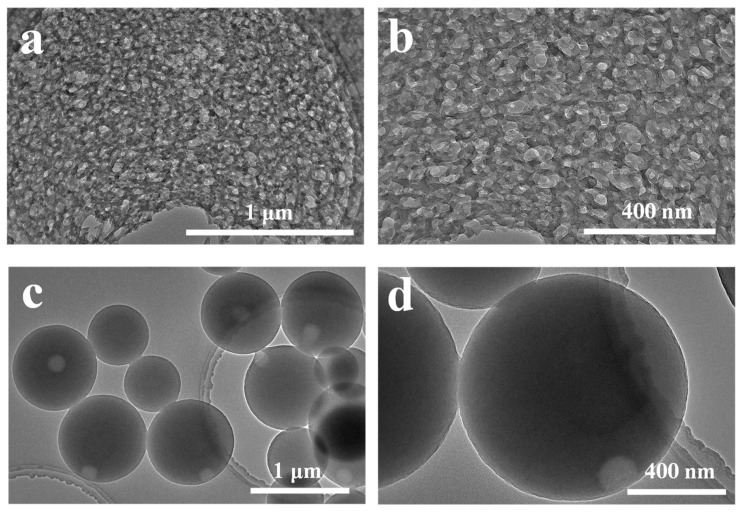
TEM images recorded for LNPs: (**a**,**b**) TLNP; (**c**,**d**) CLMP.

**Figure 8 polymers-14-04196-f008:**
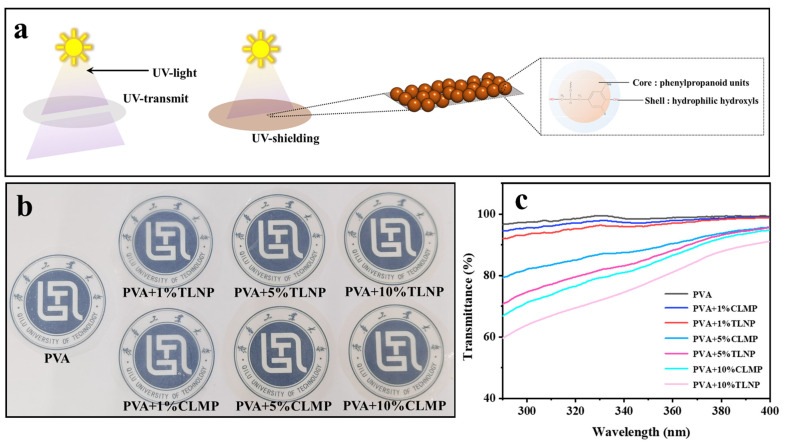
(**a**) Anti-UV mechanism diagram; (**b**) physical picture of PVA film with different lignin content; and (**c**) UV transmission diagram of different membrane materials.

## Data Availability

Not applicable.

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
