# Peer review of "Fabrication Mechanisms of Lignin Nanoparticles and Their Ultraviolet Protection Ability in PVA Composite Film"

_polymers, 2022, doi:10.3390/polym14194196_

Round 1
Reviewer 1 Report
Dear Authors,
Your study is quite interesting, but it needs to be improved in many respects.
Order the content by the typical structure of a scientific article. Example -l. 104-131. Is it a method description, summary, or abstract? Why do you write what and how you proceed in the introduction?
Please let me write and understand the aim of your study. Is it specified anywhere?
Explain all abbreviations.
Improve the quality of the figures. Example: Small pictures in the graphs of fig. 3.
Subchapter 3.3 is methodic, but chapter 3 is Results. Please order the content.
Fig. 5 - are there your results illustrated?
Please establish a new chapter discussion.
Results presentation -what is the meaning of red curves in fig 3?
Fig. 4 why is intensity shown in eV and a.u.?What is the scale? Weight in %? Do you expect 120%? Please explain.
What does it mean "narrow distribution"? How has narrowness been measured? It's not science. Only speculations. Maby it would help if you calculated the kurtosis of the distribution or other characteristics of the distribution.
Please prepare your conclusion according to the aim of the study and generalize your results. 402-409 speculations not based on your results.
401-402. How can you relate? This is the question, I think.
Reviewer 2 Report
Manuscript Number: polymers-1940423
Title: Fabrication Mechanisms of Lignin Nanoparticles and Their Ultraviolet Protection Ability in PVA Composite Film.
Article Type: Research Paper
In the manuscript the experimental research concerning synthesis of lignin nanoparticles is presented. Authors investigate the mechanism of sphere formation. Moreover a special PVA films were created in order to research the UV absorption properties of received product. The manuscript is well written. In my opinion everything is clearly presented. The biggest advantage of the paper is the analysis of sphere formation on the basis of crystallization and nucleation theory. The weakest part of the manuscript is the very short analysis of anti UV properties of the product. Below I am presenting my remarks:
1. Text editing should be corrected. Sometimes the references are given like [32,33] and sometimes [37][38]. Some minor errors should be corrected like” it is “et al” it should be “et al.”. After some references dots appear, for example “Liu et al [29]. Prepared”.
2. Line 40: Are you absolutely sure that the black liquor is discharged directly to the atmosphere? Perhaps, Authors had in mind rivers and lakes?
3. Line 134: “Kraft lignin (KL) was provided by Rizhao Huatai Paper Co., Ltd., and it was used with further purification.”
whereas
Line 141: “KL was purified based on the protocols reported in our previous work [45].”
So, it was purified or not?
4. Line 198: “Figures 2a, b, c, and d corresponded to the samples that were dialyzed for 2 h, 4 h, 8 h, and 16 h, respectively, and Figures 2e, f, g, and h corresponded to the samples that were dialyzed for 2 h, 4 h, 8 200 h, and 16 h, respectively.” This sentence has no sense.
5. Figure 2. Please correct the caption. Write clearly that the first row corresponds to the TLNP and the second row corresponds to CLMP. Explain, what does the X and Y values mean (dialysis time).
6. Please give more information about the determination of PSDs. What kind of size is presented in Fig. 3 (what kind of size (type of diameter) is used by the measurement software)? On the basis of how many independent images the PSDs were determined. How big was the population of particles (number of records in database used for determination of PSD) It would be beneficial if Authors measure the PSD using DLS equipment.
7. Question concerning the graphs in Fig. 3: What does red lines represent? What kind of average is presented in Fig. 3 i.
8. Line 235: “According to literature reports, the absorption peaks at 2940 cm−1”. Please give the references.
9. TGA: Could Authors give an explanation in the text of the manuscript about the following issue: Why does the graphs for kraft lignin and CLMP are similar whereas the graph for TLNP differs significantly?
10. Line 287: “It can be seen that the ball formation process of TLNP is rapid and dominated by the process of ball formation, while the ball formation process of CLMP is dominated by ball growth.” This sentence has to be written again.
11. Line 319, 320: A reference is needed for that statement.
Round 2
Reviewer 1 Report
Dear Authors,
thank you for the improved version of the manuscript.
I still have some questions about your study. I'm trying to understand your intention and concept. It has to be based on a clear aim. Unfortunately, I can't find it in your manuscript. Also after revision in the last paragraph of the Intro. It sounds pretty as a summary. Please describe in a clear way what you will prove in your study. Please let me understand what will you get in your study.
You wrote: "We 100 aim to develop a simple, economical, high-yield, and environmentally-friendly method to 101 prepare lignin nanoparticles."
But you didn't describe how you measure these charcteristics?
In which way will you evaluate the simplicity of the developed method?
In which way will you evaluate the economic value of this method?
What about yield efficiency and environmental friendliness?
You have to prove that the developed method is simpler, more economical, more yield efficient, and more environmentally friendly than oter methods.
Please show the quantitative proofs of these facts. Then conclude it the proper way.
Good luck.
Round 3
Reviewer 1 Report
Dear Authors,
thank you for your explanations.
I hope the content of the manuscript will be ten more intelligible to readers.
Best regards